# Type-I hyperbolic metasurfaces for highly-squeezed designer polaritons with negative group velocity

Yihao Yang[1,2,3,4,6], Pengfei Qin[2,6], Xiao Lin [3], Erping Li [2], Zuojia Wang [5], Baile Zhang [3,4] & Hongsheng Chen [1,2]

Hyperbolic polaritons in van der Waals materials and metamaterial heterostructures provide unprecedented control over light-matter interaction at extreme nanoscales. Here we propose a concept of type-I hyperbolic metasurface supporting highly-squeezed magnetic designer polaritons, which act as magnetic analogs of hyperbolic polaritons in the hexagonal boron nitride (h-BN) in the first Reststrahlen band. Compared with the natural h-BN, the size and spacing of the metasurface unit cell can be readily engineered, allowing for manipulating designer polaritons in frequency and space with greater flexibility. Microwave experimental measurements display a cone-like dispersion in momentum space, exhibiting an effective refractive index up to 60 and a group velocity down to $c/400$. Tailoring the metasurface, we demonstrate an ultra-compact integrated designer polariton circuit including high-transmission 90° sharp bending waveguides and waveguide splitters. The present metasurface could serve as a platform for polaritonics, and find applications in waveguiding, terahertz sensing, subdiffraction focusing/imaging, low-threshold terahertz Cherenkov radiation, and wireless energy transfer.

[1] State Key Laboratory of Modern Optical Instrumentation, College of Information Science and Electronic Engineering, Zhejiang University, Hangzhou 310027, China. [2] Key Laboratory of Advanced Micro/Nano Electronic Devices & Smart Systems of Zhejiang, The Electromagnetics Academy at Zhejiang University, Zhejiang University, Hangzhou 310027, China. [3] Division of Physics and Applied Physics, School of Physical and Mathematical Sciences, Nanyang Technological University, 21 Nanyang Link, Singapore 637371, Singapore. [4] Centre for Disruptive Photonic Technologies, The Photonics Institute, Nanyang Technological University, 50 Nanyang Avenue, Singapore 639798, Singapore. [5] School of Information Science and Engineering, Shandong University, Qingdao 266237, China. [6]These authors contributed equally: Yihao Yang, Pengfei Qin. Correspondence and requests for materials should be addressed to Z.W. (email: z.wang@sdu.edu.cn) or to B.Z. (email: blzhang@ntu.edu.sg) or to H.C. (email: hansomchen@zju.edu.cn)

Naturally hyperbolic materials that support highly-confined hyperbolic polaritons have recently emerged as an innovative platform to confine light at extreme nanoscales[1–3]. In hyperbolic materials, signs of in-plane and out-of-plane permittivity/permeability are opposite. According to the signs of the out-of-plane permittivity/permeability, the hyperbolic materials can be classified into two types: $\varepsilon_\perp$ $(\mu_\perp) < 0$ and $\varepsilon_\parallel$ $(\mu_\parallel) > 0$, for type-I hyperbolic materials; $\varepsilon_\perp$ $(\mu_\perp) > 0$ and $\varepsilon_\parallel$ $(\mu) < 0$, for type-II hyperbolic materials. Here, $\parallel$ and $\perp$ represent the in-plane and out-of-plane components of permittivity/permeability, respectively. According to the electromagnetic (EM) theory, dispersion of the type-I hyperbolic media is a two-sheeted hyperboloid, and that of the type-II hyperbolic media is a single-sheeted hyperboloid[4].

As a representative naturally hyperbolic material, polar dielectric material of hexagonal boron nitride (h-BN) supports hyperbolic phonon-polaritons at two separated Reststrahlen bands in mid-infrared regime[3,5–8]. Interestingly, its phonon polaritons in lower (760–820 cm$^{-1}$) and upper (1365–1610 cm$^{-1}$) Reststrahlen bands show type-I and type-II hyperbolic dispersions, respectively. Special attention has been given to the lower Reststrahlen band because the phonon-polaritons of layered h-BN slab have an opposite group and phase velocity in this band[8]. Besides, experimental investigations have demonstrated numerous merits of the phonon polaritons in the h-BN, such as high confinement, ultra-short wavelength, and low loss compared with metal-based surface plasmons and graphene plasmons, which makes it an excellent candidate for nano-photonics[9,10]. The h-BN holds a promising future in applications for sub-diffraction imaging[5] such as hyperlens, enhanced light–matter interaction, super-Planckian thermal emission, and so forth. However, the h-BN only works as a hyperbolic material in the narrow Reststrahlen frequency bands, beyond which no phonon-polariton exists.

Here, we propose a concept of type-I hyperbolic metasurface with anisotropic magnetic responses, to mimic the hyperbolic phonon-polaritons in h-BN and to engineer polaritons at will in frequency and space. The metasurface consists of a single-layer coil array and is characterized by a negative/positive out-of-plane/in-plane permeability. Note that the proposed metasurface is qualitatively different from the conventional hyperbolic metasurfaces, where in-plane surface plasmons[11] show hyperbolic dispersion relations and propagate with convergent manners[12–15]. The type-I hyperbolic metasurface behaves as an artificial h-BN in its first Reststrahlen band in many ways. For examples, the designer polaritons on the metasurface carry ultra-high momenta and ultra-large negative group velocities. By directly imaging near-field distributions over the metasurface at microwave frequencies, we experimentally observe a cone-like dispersion in reciprocal space, with a remarkably high effective refractive index up to 60 and a large group velocity down to $c/400$ ($c$ is the speed of light in vacuum). With so many exciting features and merits of low profile, lightweight, and ease of access, the type-I hyperbolic metasurfaces have great potentials in applications. First of all, the high effective refractive index makes the metasurface an excellent candidate of highly-integrated waveguide circuits for two reasons: (i) the high effective refractive index is beneficial to miniaturization of waveguide circuits; (ii) the designer polaritons travel smoothly through the compact sharp-corner waveguides and waveguide splitters with high transmissions, which can be explained by the quasi-static approximation[16–18]. Taking advantage of the high effective index, we experimentally achieve an entire integrated polariton circuit with a footprint shrunken by almost 3600 times, substantially exceeding the conventional waveguide circuits (typically a few times). It enables the size of the entire polariton circuit to be under the diffraction limit ($\lambda_0/2$, where $\lambda_0$ is the free-space wavelength), which is highly pursued in polaritonics (including plasmonics). Besides, the

large effective refractive index of the present metasurface could be used to design subdiffraction focusing/imaging devices[5] and low-electron-velocity terahertz Cherenkov radiation emitters[19]. Second, the significant group velocity greatly enhances the light–matter interaction and makes the metasurface extremely sensitive to the thickness and refraction index of the surroundings, and, thus, an excellent electromagnetic wave sensor. Third, due to the duality between electric and magnetic phenomena, the magnetic hyperbolic polariton with highly-squeezed modes could provide a pathway for achieving strong magnetic transition enhancement[19]. Fourth, the process of EM energy transport on the metasurface is precisely the same as the well-known wireless energy transfer[20–22]. Therefore, the present metasurface may inspire novel wireless energy transfer devices. Finally, the present metasurface could be an alternative platform in polaritonics, with tailorable dispersions in frequency and space.

## Results

**Design of type-I hyperbolic metasurfaces.** The proposed type-I hyperbolic metasurface consists of arrays of coiling copper wires patterned on a dielectric substrate (Fig. 1a). The figure inset shows the coil details, where the golden and brown regions denote a coppers layer and a dielectric substrate, respectively. Here, $a = 2$ mm; $w = g = 0.2$ mm; numbers of turns $N = 15$; $p = 13.8$ mm; $t = 1$ mm. The thickness of the metal layer is 0.035 mm; the conductivity of copper is $5.7 \times 10^7$ S m$^{-1}$; the relative permittivity of the substrate is $2.55 + 0.001i$ below 10.0 GHz. Besides, the metasurface is extremely thin (around $1/10^3$ times of the operational wavelength).

By employing eigenvalue module of a commercial software Computer Simulation Technology (CST) Microwave Studio, we obtain iso-frequency contours (IFCs) in the first Brillouin zone (FBZ) of the fundamental mode on the present metasurface (Fig. 1b). Intriguingly, the dispersion shows a cone-like topology, rather than an inverted cone as usual cases. Because group velocity $\mathbf{v}_g$ is calculated with[23]

$$\mathbf{v}_g = \partial \omega / \partial \mathbf{k} \qquad (1)$$

where $\omega$ and $\mathbf{k}$ are angular frequency and wavevector, respectively. The group velocity of the designer polaritons on the metasurface is negative. The eigenmodes on the metasurface are shown in Fig. 1c, where the designer polaritons are highly confined both vertically and horizontally. The magnetic field distribution indicates a strong magnetic dipole, which physically arises from the surface currents flowing along the spiral coil.

**Effective constitutive parameters.** To understand the exotic behavior of the designer polaritons, we construct a layered metamaterial by stacking the metasurface periodically along $z$-direction, as shown in Fig. 2a. As the metal coils produce $z$-oriented magnetic resonances, this artificial material works as a type-I magnetic hyperbolic metamaterial. By applying a well-established retrieval process[24], we obtain effective constitutive parameters of the constructed metamaterial (Fig. 2b), where $\mu_z$ is negative from 0.315 to 0.4 GHz, while the other constitutive parameters are positive. Then we study EM properties of a metamaterial slab with a finite thickness in the $z$-direction. Considering the fundamental transverse electric (TE) even mode (the inset of Fig. 2c), the corresponding dispersion of the metamaterial slab is

$$\mathbf{k}_d - \mathbf{k}_m/\mu_y \times \tanh(\mathbf{k}_m \times d/2) = 0,$$
$$\text{with } \mathbf{k}_d = \sqrt{\boldsymbol{\beta}^2 - \mathbf{k}_0^2}, \mathbf{k}_m = \sqrt{(\boldsymbol{\beta}^2/\mu_z - \mathbf{k}_0^2 \varepsilon_x)\mu_y}, \qquad (2)$$

where $d$ is the thickness of the metamaterial slab, $\boldsymbol{\beta}$ represents the wavevector of the designer polaritons along the propagating

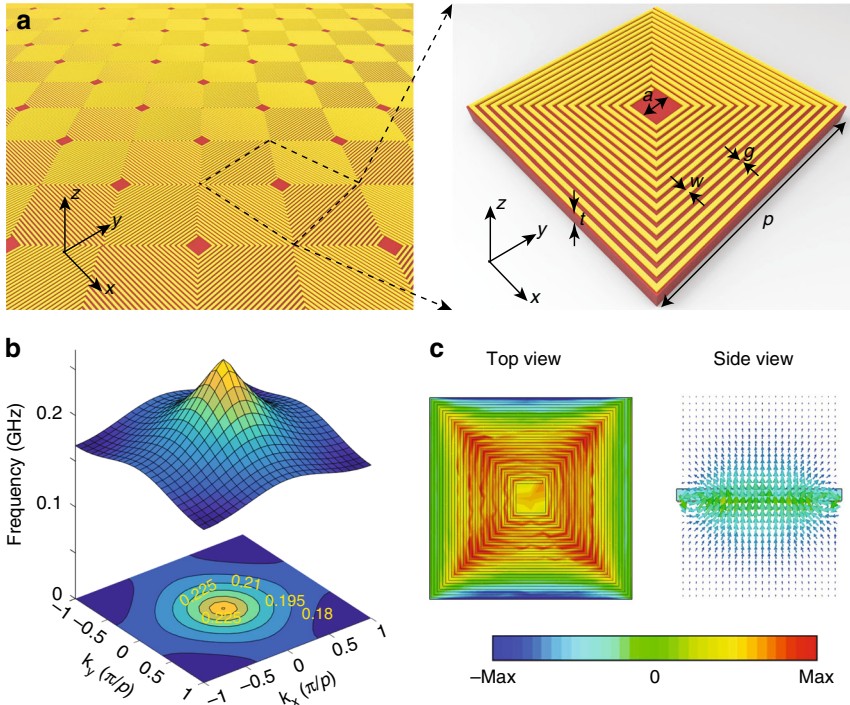

**Fig. 1** Type-I hyperbolic metasurface. **a** Type-I hyperbolic metasurface consisting of coiling copper wires patterned on a dielectric substrate. The inset shows the unit cell details. Here, $a = 2$ mm; $w = g = 0.2$ mm; numbers of turns $N = 15$; $p = 13.8$ mm; and $t = 1$ mm. The thickness of the metal layer is 0.035 mm; the conductivity of copper is $5.7 \times 10^7$ S m$^{-1}$; the relative permittivity of the substrate is $2.55 + 0.001i$ at 10 GHz. **b** Isofrequency contour of the dispersion in the first Brillouin zone. **c** Side and top view of the magnetic field distribution of the designer polariton, respectively. The color bar measures the magnetic field intensity

direction, and $\mathbf{k}_0$ denotes the wavevector in vacuum. By substituting the retrieval constitutive parameters to Eq. (2), we can obtain the dispersions of metamaterial slabs with different thicknesses. Here, the metamaterial slabs with 1, 2, 3, and 4 layers are considered. After choosing proper effective thicknesses, we find that the calculated dispersions match with the simulated counterparts excellently. Note that as the numbers of layers are small, edge effects slightly change the EM properties of the metamaterial slab. Therefore we use the effective thicknesses rather than physical thicknesses. Besides, we visualize the dispersion of the surface plasmons via a false-color plot of |Im (ref)|, where ref is the reflection coefficient of TE wave[25]. This is because the designer polaritons propagating in the metamaterial slab are the singularity poles in the reflection coefficient. Interestingly, Eq. (2) holds even for the metamaterial slab with a single layer, namely a metasurface, which manifests that the z-oriented negative magnetic response mainly arises from the self-induced inductances and capacitances inside the coils rather than the interlayer coupling.

**Dependence of designer polaritons on metasurface parameters**. In the following, we study the performance dependence of the designer polaritons on the metasurface parameters. When the substrate thicknesses are smaller than 4 mm, altering the thicknesses will dramatically change the dispersions, which manifests that the EM responses of the metasurface are very sensitive to the thicknesses (Fig. 3a). When the thicknesses are larger than 4 mm, the dispersions almost stay the same, indicating that the designer polaritons are highly confined around the metasurface and dramatically decay into the background (Fig. 3a). Besides, the metasurface dispersions are very sensitive to the substrate permittivity (Fig. 3b). Such a remarkable sensitivity to the substrate thicknesses and permittivity may find applications in terahertz sensing[26,27].

We also study the relations between the number of turns and the maximal squeezing factor. Here, the squeezing factor or the effective refractive index is defined as $\boldsymbol{\beta}/\mathbf{k}_0$. One can see that the maximal squeezing factor is almost proportional to the number of turns (Fig. 3c). In naturally hyperbolic materials, an increased squeezing factor always associates with a higher group velocity, and thus a severer propagation loss, which imposes a limit on the largest squeezing factor[10]. However, for the present metasurface, removing the lossy substrate and using the superconductor coils[28], the ohmic loss is almost neglectable. Therefore, there is no limitation for the largest squeezing factor. Finally, we change the period and keep the other parameters the same. One can see that when enlarging the period, the bandwidth decreases and the group velocity increases (Fig. 3d). This is because the coupling between neighbor unit cells diminishes as the period increases, and the energy can only transmit via a strong magnetic resonance of the coils. The limiting case is that the period is so large that the whole band becomes flat with a narrow bandwidth centered at the magnetic resonance frequency of a single isolated coil. In that case, the physical mechanism of the energy transfer between two neighbor coils is the so-called wireless energy transfer[21].

**Measured dispersion of type-I hyperbolic metasurface**. We have carried out several experiments to characterize the proposed metasurface. The fabricated sample consists of $22 \times 22$ unit cells, taking up an area of 300 mm × 300 mm (see Fig. 4b). In the experiments, a port of vector network analyzer (VNA), as a broadband source ($0^+$ to 0.5 GHz), directly connects to a coil unit cell at the metasurface edge, to drive the electrons in the coil. This is a high-efficiency way to excite the designer polaritons in our system. The detector, a compact coil antenna with magnetic resonance around 0.2 GHz, is fixed at an arm of a three-dimensional

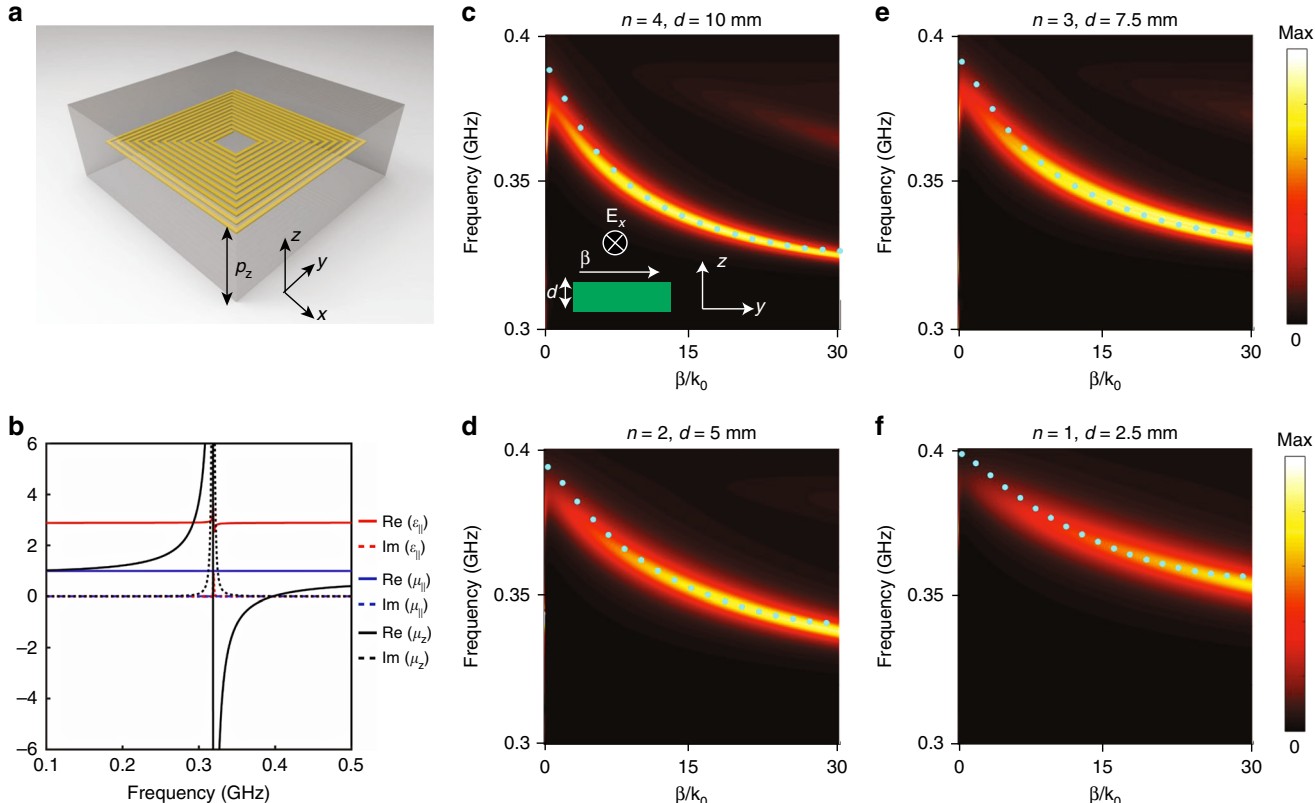

**Fig. 2** From type-I hyperbolic metamaterials to type-I hyperbolic metasurfaces. **a** Scheme of the hyperbolic metamaterials composed of coiling copper wires. Here, $p_z = 5$ mm; $n = 12$; $p = 13.8$ mm; $w = g = 0.2$ mm; and $a = 2$ mm. The coiling metal wires are embedded in a dielectric host with relative permittivity 2.55. **b** Retrieved constitutive parameters of the hyperbolic metamaterial. The hyperbolic region is from 0.315 to 0.4 GHz. $\varepsilon_\parallel$, $\mu_\parallel$, and $\mu_z$ are the in-plane permittivity, in-plane permeability, and out-of-plane relative permeability, respectively. **c–f** Dispersions of the metamaterial slabs with 4, 3, 2, and 1 layer, respectively. The green dots are the dispersion of the practical metamaterial structures. The false-color plots are the dispersion relations of the metamaterial slabs with the retrieval constitutive parameters. Here, $d$ is the effective thickness of the metamaterial slab

movement platform. Both the source and the detector are connected to the VNA to get the amplitude and phase of the measured magnetic field. With the above system, we scan the $z$-oriented magnetic field distributions on the plane 5 mm above the metasurface, as shown in the right column of Fig. 4a. Note that the field pattern at 0.198 GHz shows two privileged directions, due to the "distorted" isofrequency contour near the Brillouin zone boundaries (see Fig. 1b). Then we apply the spatial Fourier transform to the measured complex field patterns and obtain the momentum space (the left column of Fig. 4a).

Our experimental results display a cone in momentum space, consistent with the theoretical prediction (Fig. 1b). One can see that the isofrequency contours are almost circular. This is because we do not use the absorbers to prevent the reflections at metasurface edges, which provide momenta of $\mathbf{k}_x < 0$. Besides, the spots at the FBZ center represent the radiative noise whose energy is relatively small, comparing with that of the designer polaritons. Counterintuitively, we directly observe that the wavelength becomes longer as the frequency increases. We also retrieve the squeezing factor from the experimental data. Impressively, the squeezing factor ranges from 8 up to 60, higher than typical values in many previous metamaterials and metasurfaces[11,24,29–31]. Such a large squeezing factor is comparable or even exceed that in the two-dimensional materials, such as graphene plasmons[10]. We should note that with more turns of the coils, the squeezing factor can be even larger and has no strict limits. Such a sizeable squeezing factor is highly beneficial in miniaturization of the integrated waveguide circuits. Here, the squeezing factor is as large as 60, which means we can use it to

shrink the footprints of integrated waveguide circuits by almost 3600 times, substantially exceeding the conventional waveguide circuits (typically a few times). Besides, we also retrieve the group velocity of the designer polaritons from the fitted dispersion by applying Eq. (1), which ranges from $3 \times 10^6$ m s$^{-1}$ to $7.5 \times 10^5$ m s$^{-1}$, or from $0.01c$ to $0.0025c$. Therefore, such a metasurface shows an ultraslow light effect in a broad band (a relative bandwidth of 32.6%). The prominent performance of the metasurface may lead to numerical applications, such as delay lines, EM energy storage, and strongly enhanced light–matter interaction.

**A deep sub-wavelength integrated waveguide circuit.** As the type-I hyperbolic metasurface shows so many fascinating properties, it could find plenty of applications. As an example, by taking advantages of the high refraction index of the type-I hyperbolic metasurface, we design an ultra-compact integrated waveguide circuit. We first tailor the metasurface to meta-ribbons with a single-unit-cell width. With the meta-ribbons, we construct a waveguide splitter with several 90° sharply twisted corners (Fig. 5b). From the measured magnetic field distribution over the waveguide circuit (Fig. 5c–h), one can see that the designer polaritons are launched at the excitation, split into two beams at the splitter, and smoothly pass through the 90° sharp corners of the waveguide. We note that such highly-squeezed modes with negative group velocity are unique to the type-I hyperbolic material slab, and have not been observed in previous surface-wave waveguides[11,29,30,32,33]. Besides, the meta-ribbon exhibits the same dispersion of the type-I hyperbolic metasurface (see Supplementary Note 1). Therefore, the waveguiding

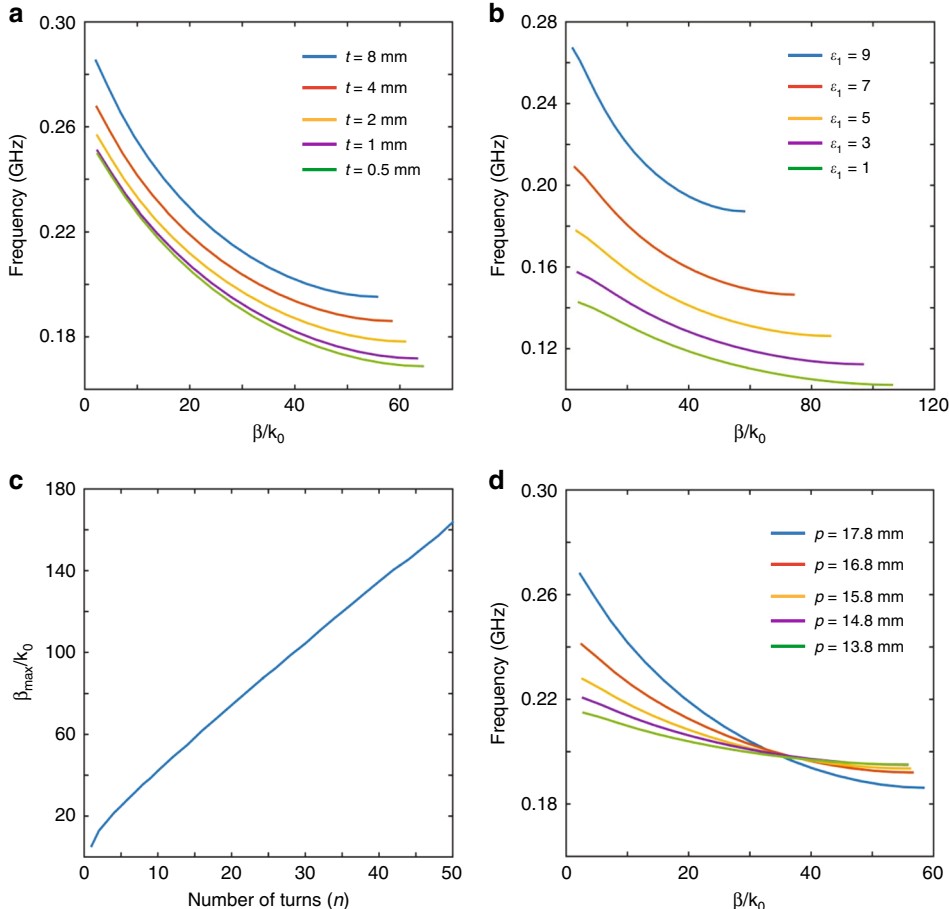

**Fig. 3** Dependence of designer polaritons on metasurface parameters. **a** Dispersions of the metasurfaces with different substrate thicknesses. **b** Dispersions of the metasurfaces with different substrate permittivity. **c** Maximum squeezing factor as a function of numbers of turns. Here, we alter the numbers of turns while maintaining the other parameters (except the period) the same. Therefore, the period becomes longer as the number of turns increases. **d** Dispersions of the metasurfaces with different periods. The coils keep the same and the distance between neighbor coils changes

effect of the meta-ribbon is directly related to the type-I hyperbolic metasurface.

Interestingly, the transmission of designer polaritons through the splitter and sharp corners are very high. The underlying reason is the small effective width of the meta-ribbon in comparison with the effective wavelength of the designer polaritons, and thus we can apply a common quasi-static approximation[16–18]. With this approximation, the sharp-corner waveguide can be viewed as a junction with two transmission lines with the same impedance, and the splitter can be viewed as a junction with one input transmission line and two output transmission lines with the same impedance. Therefore, the reflection loss of the bending waveguide is negligible, and that of the splitter is about 12%. Note that such a high transmission has also been found in photonic crystal waveguides[34,35] and plasmonic waveguides[16–18]. Besides, one can see that the meta-ribbons work excellently from 0.193 to 0.217 GHz. We emphasize that the whole structure size is only about $1/4 \times 1/4$ free-space wavelength. Therefore, we experimentally achieve an ultra-compact integrated designer polariton circuit beyond the diffraction limit.

## Discussion
Our work thus identifies a class of hyperbolic metasurfaces, namely type-I hyperbolic metasurfaces, which behaves in many ways the same as an artificial h-BN in the h-BN's first Reststrahlen band, such as extremely high squeezing factors and ultra-large negative group velocities. In comparison with the natural h-BN, the artificial type-I

hyperbolic metasurface is readily geometry-tailorable, allowing for the creation of designer polaritons with almost arbitrary dispersions in both frequency and space. The present metasurface with a low profile, lightweight, and ease of access, could serve as an alternative platform in polaritonics and may find many other potential applications, such as electromagnetic wave sensors, subdiffraction focusing/imaging, low-electron-velocity terahertz Cherenkov radiation emission, strong magnetic transition enhancement, and wireless energy transfer. In combination with flexible substrates[30] and active and nonlinear components[36,37], we envision further exciting possibilities such as actively controlling conformal designer polaritons with nonlinear properties at deep subwavelength scales.

Although demonstrated at microwave frequencies, the concept of type-I hyperbolic metasurface is general and applicable to frequencies up to far-infrared regime (see a design of a far-infrared type-I hyperbolic metasurface in Supplementary Note 2). We also notice that at higher frequencies, metallic losses become considerable and cannot be neglected. Therefore, when increasing the maximal squeezing factor, the figure of merit ($\mathrm{Re}(k)/\mathrm{Im}(k)$) decreases, imposing a limit on the maximal squeezing factor. To overcome this challenge, one may use gain medium[38] to compensate for the dissipative losses, which needs future investigation.

## Data availability
The data that support the plots within this paper and other findings of this study are available from the corresponding author upon request.

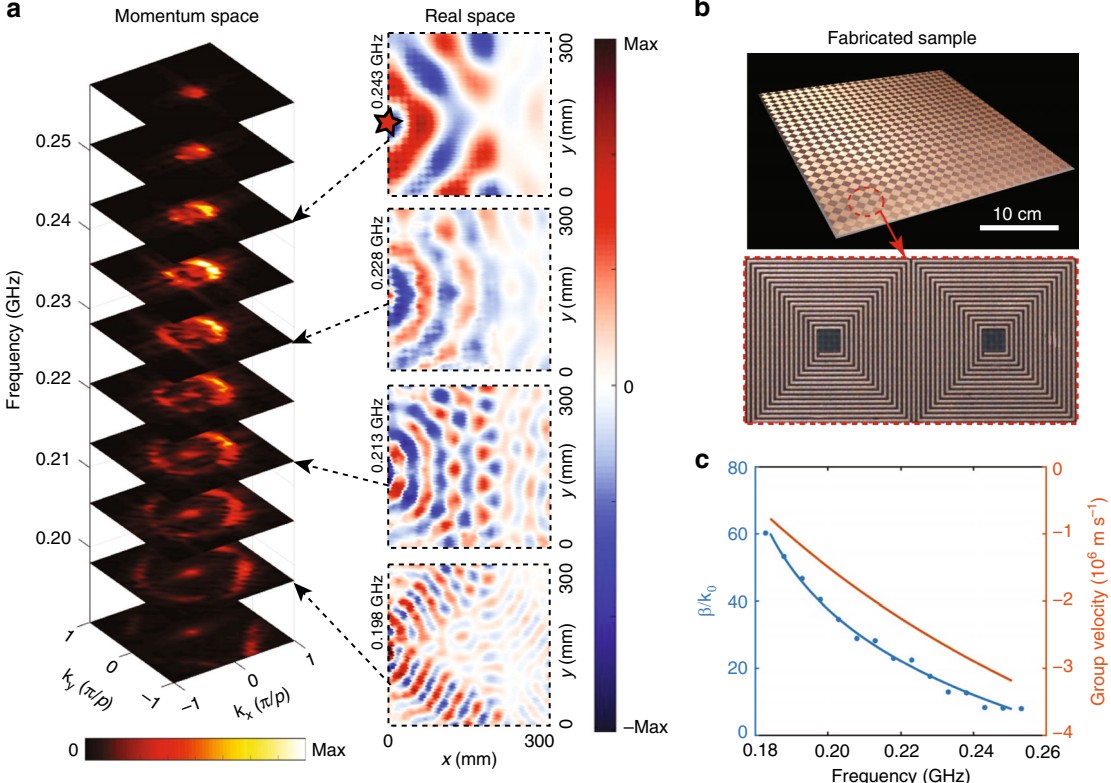

**Fig. 4** Measured momentum space, field distributions in real space, and retrieved dispersion and group velocity of the designer polaritons on the type-I hyperbolic metasurface. **a** Measured momentum space of the designer polaritons in the FBZ, and measured $Hz$ field distributions on the plane 5 mm over the metasurface (22 by 22 unit cells) at 0.198, 0.213, 0.228, and 0.243 GHz, respectively. The red star in the right inset represents the source location. **b** Photograph of the metasurface sample. Inset: two unit cells of the metasurface. **c** Retrieved squeezing factor and group velocity of the designer polaritons from the experimental results. Here, the blue dots are the experimental data; the blue curve represents exponential fits to the experimental data; the orange curve is the group velocity obtained from the blue curve

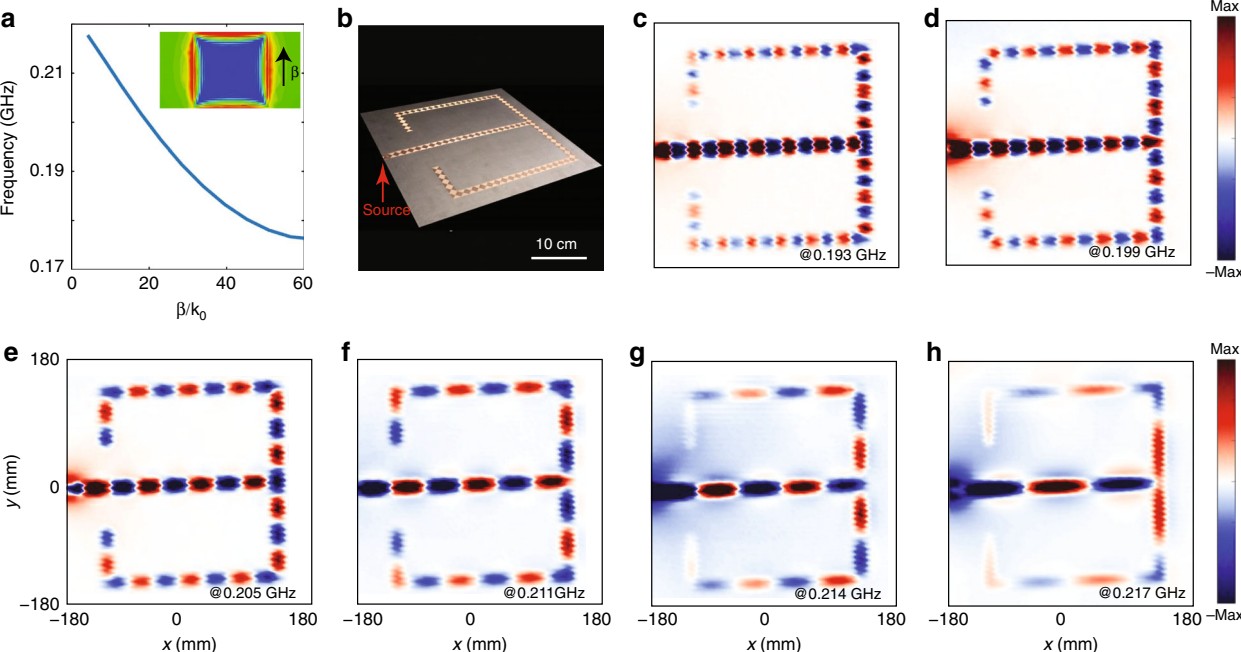

**Fig. 5** A deep sub-wavelength integrated waveguide circuit based on the type-I hyperbolic metasurface. **a** Dispersion of the meta-ribbon. **b** A fabricated sample of the deep-sub-wavelength waveguide circuit. The red arrow represents the source location. **c–h** Measured $Hz$ field distributions on the plane 5 mm over the metasurface at 0.193 GHz (**c**), 0.199 GHz (**d**), 0.205 GHz (**e**), 0.211 GHz (**f**), 0.211 GHz (**g**), and 0.217 GHz (**h**), respectively. Here, the whole measured region is below the diffraction limit, i.e., half of the free-space wavelength

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

## Acknowledgements

Work at Zhejiang University was sponsored by the National Natural Science Foundation of China under Grant Nos. 61625502 and 61574127, the Top-Notch Young Talents Program of China, and the Innovation Joint Research Center for Cyber-Physical-Society System. Work at Nanyang Technological University was sponsored by Singapore Ministry of Education under Grant Nos. MOE2018-T2-1-022 (S), MOE2015-T2-1-070, MOE2016-T3-1-006, and Tier 1 RG174/16 (S).

## Author contributions

Y.Y. conceived the original idea and designed the metasurface. P.Q. and Y.Y. carried out the experiments. Y.Y., Z.W., X.L., E.L., B.Z. and H.C. produced the manuscript and interpreted the results. Y.Y., Z.W., B.Z. and H.C. supervised the project. All authors participated in discussions and reviewed the manuscript.

## Additional information

**Competing interests:** The authors declare no competing interests.

