## [Peer Review File · Nature Communications]

Reviewers' comments:

Reviewer #1 (Remarks to the Author):

This is an interesting manuscript, which describes hyperbolic metasurfaces exhibiting highly-squeezed designer polaritons, which might potentially be accepted for publication in Nature Communications. However, as it stands, it requires revision.

1) Some figures are not clear. For example, Fig.4(a) consists of two parts which relationship to each other (indicated by arrows) is not obvious. Moreover, the scale of (theoretical?) images on the right hand side is not shown.

2) The waveguide demonstration in Fig.5 looks interesting. However, it is not clear if the designed properties of the hyperbolic metasurface play any role in the waveguiding effect. Since the guided mode is tightly localized, the same behavior may probably be realized in an appropriately designed one dimensional waveguide in the absence of the metasurface itself.

3) The English of the manuscript needs improvement.

Reviewer #2 (Remarks to the Author):

In the paper titled « Type-I hyperbolic metasurfaces for highly-squeezed designer polaritons with negative group velocity » from Yihao Yang et al., the authors design a type-I hyperbolic metasurface, with a copper coil unit cell, supporting magnetic designer polaritons. The highly-confined designer polaritons are studied in the microwave regime: they demonstrate negative group velocity and they suffer from negligible losses. The theoretical studies are accompanied by experimental results, corroborating the dispersion obtained theoretically. Finally, the authors realize an integrated waveguide circuit constituted of a beam splitter with 90° sharp corners with high-transmission.

To the best of my knowledge, the proposed study is original and interesting, offering a new tool to engineer at will low-loss polariton dispersion, with negative group velocities. I think that those results will be useful and beneficial for the broad optical community, if the authors wave the confusion between the microwave range and the optical range.

1) My major concern is about the losses and the microwave range. The authors are quite evasive in the abstract and in the introduction that the frequency range under study is the microwave range, making it sound that the current study is performed in the optical range. Moreover, all the applications cited and their references are for the optical or infrared range: integrated waveguide

circuits (line 73), imaging devices (line 82), Cherenkov radiation emitters (line 82), optical sensors (line 85) and magnetic transitions of emitters (line 87).

Therefore, in order to justify those considerations, a discussion on the properties (mainly the losses, but also the smallest feature size) of the designer polariton in the optical or infrared range would be interesting, especially since copper behaves almost like a perfect metal in the microwave regime (Nature Photonics volume 2, pages 175–179, 2008). The losses limit the hyperbolic dispersion and therefore the high effective refractive index obtained (as you discuss in lines 178-180), maybe reducing the utility of this interesting design for the cited applications.

2) On the same note, on line 213, you compare your high factor ranges from 8 to 60 to ordinary metamaterial and metasurfaces. What are the usual squeezed factor ranges of ordinary metamaterial and metasurfaces? In the ref. 2 you cite, I don't see such discussion.

3) I am not an experimentalist and I think that the paper would be greatly improved with more details on the experimental section, discussed in line 200 to 208.

First, Fig. 4 needs more information. In Fig. 4a, the Hz field scans do not have x and y scale, so I don't understand if it is a scan of the total metasurface, only a unit cell, or X unit cells. Looking at the dispersion, I would have expected a circular field pattern, but there seem to be two privileged directions (in the low frequency Hz scan): could you comment those results? Also, are there reflections on the edge of the metasurface or is the designer polariton absorbed before reaching the edge? Or do we excite various frequencies with the incoming port? Could you describe in more details how you excite exactly the designer polariton? Is there a part of the excitation that is not coupled with it and creates radiative noise? Fig. 4b is too small to see whether it is fabricated or not, and there is no scale bar. Maybe you could provide a zoom of one unit cell?

Typos:

- Lines 40 and 41: the "parallel" sign is not there.
- Line 49: change Especial to Special
- Subfigures of Fig. 3 are not lettered.
- Lines 175-176 : is the number of turns increased for fixed period?
- Figure 4 legend: orange curve of the group velocity, not yellow.
- Line 207, 5 nm or 2 mm?

Response Letter to Reviewers

We are grateful for the constructive comments on this manuscript (NCOMMS-18-20953717-T) from all the reviewers.

In the text below each of the comments from each reviewer is quoted in italics and is followed by the corresponding detailed response. We also revised the manuscript and the Supplementary Information accordingly, and these updates are highlighted in blue and by a vertical red line in the left margin in those files. In the text below the references to these updates are highlighted in a similar way (i.e., by a vertical red line).

GENERAL COMMENTS FROM 1st REVIEWER:

This is an interesting manuscript, which describes hyperbolic metasurfaces exhibiting highly-squeezed designer polaritons, which might potentially be accepted for publication in Nature Communications. However, as it stands, it requires revision.

Response from Authors:

We thank the reviewer for the positive comments and considering that “*This is an interesting manuscript, ..., which might potentially be accepted for publication in Nature Communications.*” In the following, we fully address the specific comments point-by-point.

SPECIFIC COMMENTS FROM 1st REVIEWER:

1st Reviewer -- Comment 1:

Some figures are not clear. For example, Fig.4(a) consists of two parts which relationship to each other (indicated by arrows) is not obvious. Moreover, the scale of (theoretical?) images on the right hand side is not shown.

Response from Authors:

We thank the reviewer for pointing out this issue. Following the reviewer’s suggestion, we have improved the figures. In Fig. 4(a), the left and right parts are the momentum space and the real space, respectively, of the measured electromagnetic fields, which are linked by Fourier transform. We have added annotations in each subfigure. Besides, following the reviewer’s suggestion, we have added the scale of the image on the right-hand side.

1st Reviewer -- Comment 2:

The waveguide demonstration in Fig.5 looks interesting. However, it is not clear if the designed properties of the hyperbolic metasurface play any role in the waveguiding effect. Since the guided mode is tightly localized, the same behavior may probably be realized in an appropriately designed one dimensional waveguide in the absence of the metasurface itself.

Response from Authors:

We thank the reviewer for considering that “*the waveguide demonstration in Fig.5 looks interesting.*”

The reviewer is certainly correct that “*an appropriately designed one dimensional waveguide in the absence of the metasurface itself*” can exhibit some waveguiding behaviors similar to our demonstration in Fig. 5. For example, reference (*Phys. Rev. Applied* **9**, 044019 (2018)) shows

similar beam splitting and sharp bending of localized surface modes. This, as the reviewer reasoned, is because “*the guided mode is tightly localized*”.

However, we would like to clarify that Fig. 5 demonstrates not only that “*the guided mode is tightly localized*”, but also that the guided modes propagate with negative group velocities (i.e., the designer polaritons with a larger momentum appear at a lower frequency). Such highly-squeezed modes with negative group velocities in a broad band are unique to the type-I hyperbolic material slab, and have not been observed in previous surface-wave waveguides (*Science* **305**, 847-848 (2004); *Phys. Rev. Lett.* **97**, 176805 (2006); *Nature Photon.* **2**, 175-179 (2008); *Proc. Natl. Acad. Sci.* **110**, 40-45 (2013); *Phys. Rev. Applied* **9**, 044019 (2018)). That is why reviewer #2 commented that our work offers “*a new tool to engineer at will low-loss polariton dispersion, with negative group velocities*”. Note that similar guided modes with negative group velocities can exist in a thin hexagonal boron nitride (*h*-BN) slab, because the *h*-BN is also a type-I hyperbolic material in the first Reststrahlen band (*Nat. Commun.* **6**, 7507 (2015); *Nature Photon.* **9**, 674-678 (2015)). Our work serves as a metasurface analogue of the *h*-BN in the first Reststrahlen band.

To further lift the concern of the reviewer, we have done extra simulations to show that the meta-ribbon exhibits the same dispersion of a type-I hyperbolic metasurface (see “Section 1: From the type-I hyperbolic metasurface to the meta-ribbon” in the Supplementary Information). Therefore, the waveguiding effect of the meta-ribbon is directly related to the type-I hyperbolic metasurface.

To make it clear, in the main text, on page 12, line 243, we have stated that “We note that such highly-squeezed modes with negative group velocities are unique to the type-I hyperbolic material slab, and have not been observed in previous surface-wave waveguides^{11, 30, 31, 33, 34}. Besides, the meta-ribbon exhibits the same dispersion of the type-I hyperbolic metasurface (See the Supplementary Information). Therefore, the waveguiding effect of the meta-ribbon is directly related to the type-I hyperbolic metasurface.”

1st Reviewer -- Comment 3:

The English of the manuscript needs improvement.

Response from Authors:

Following the reviewer’s suggestion, we have improved the English of the manuscript.

GENERAL COMMENTS FROM 2nd REVIEWER:

In the paper titled « Type-I hyperbolic metasurfaces for highly-squeezed designer polaritons with negative group velocity » from Yihao Yang et al., the authors design a type-I hyperbolic metasurface, with a copper coil unit cell, supporting magnetic designer polaritons. The highly-confined designer polaritons are studied in the microwave regime: they demonstrate negative group velocity and they suffer from negligible losses. The theoretical studies are accompanied by experimental results, corroborating the dispersion obtained theoretically. Finally, the authors realize an integrated waveguide circuit constituted of a beam splitter with 90° sharp corners with high-transmission.

To the best of my knowledge, the proposed study is original and interesting, offering a new tool to engineer at will low-loss polariton dispersion, with negative group velocities. I think that those results will be useful and beneficial for the broad optical community, if the authors wave the confusion between the microwave range and the optical range.

Response from Authors:

We thank the reviewer for the positive comments and considering that “to the best of my knowledge, the proposed study is original and interesting, offering a new tool to engineer at will low-loss polariton dispersion, with negative group velocities.” In the following, we fully address the specific comments point-by-point.

SPECIFIC COMMENTS FROM 2nd REVIEWER:

2nd Reviewer -- Comment 1:

My major concern is about the losses and the microwave range. The authors are quite evasive in the abstract and in the introduction that the frequency range under study is the microwave range, making it sound that the current study is performed in the optical range. Moreover, all the applications cited and their references are for the optical or infrared range: integrated waveguide circuits (line 73), imaging devices (line 82), Cherenkov radiation emitters (line 82), optical sensors (line 85) and magnetic transitions of emitters (line 87).

Therefore, in order to justify those considerations, a discussion on the properties (mainly the losses, but also the smallest feature size) of the designer polariton in the optical or infrared range would be interesting, especially since copper behaves almost like a perfect metal in the microwave regime (Nature Photonics volume 2, pages 175–179, 2008). The losses limit the hyperbolic dispersion and therefore the high effective refractive index obtained (as you discuss in lines 178-180), maybe reducing the utility of this interesting design for the cited applications.

Response from Authors:

We thank the reviewer for comments about loss.

First of all, following the reviewer’s suggestion, we have explicitly stated in the abstract that our experiments are performed at microwave frequencies, as follows:

“Experimental measurements at microwave frequencies display the cone-like hyperbolic dispersion in the momentum space, associating with an effective refractive index up to 60 and a group velocity down to 1/400 of the light speed in vacuum.”

and

“Although demonstrated at microwave frequencies, the concept of type-I hyperbolic metasurface is general and applicable to higher frequencies such as the far-infrared regime.”

We also have emphasized this point in the introduction, on page 4, line 69, as follows:

“By directly imaging the near-field distribution at microwave frequencies, we experimentally observe a cone-like dispersion in reciprocal space,”

Secondly, to remove the reviewer’s concern, we have performed simulations to verify the feasibility of type-I hyperbolic metasurface in the far-infrared regime. The modern nanofabrication technology can routinely fabricate a feature size around 100 nm. Based on this feature size, we have designed a far-infrared type-I hyperbolic metasurface as shown in Fig. R1(a), where the yellow region is silver. In the simulation, the silver is described by the standard Drude model

$$\epsilon(\omega) = 1 - \frac{\omega_p^2}{\omega(\omega + i\nu)},$$

with a plasma frequency $\omega_p = 1.37 \times 10^{16} \text{ s}^{-1}$ and a collision frequency $\nu = 3 \times 10^{13} \text{ s}^{-1}$. The dispersion of the designer polaritons on the metasurface is shown in Fig. R1(c). The operational frequency is from 2.65 THz to 3.4 THz, which is in the regime of far-infrared frequency (0.3 THz to 20 THz). Besides, the figure of merit ($\text{Re}(k)/\text{Im}(k)$; FOM, for short) varies in the range from

19.2 to 22, being comparable with that of *h*-BN-encapsulated graphene (FOM~25; *Nature Mater.* **16**, 182-194 (2017)). The squeezing factor (k/k_0) reaches up to 20.

As for that the losses may affect the high effective refractive index, we also perform additional simulations as shown in Fig. R1(d). When increasing the number of turns, the maximal squeezing factor increases while the FOM decreases. Therefore, there is a trade-off between the maximal squeezing factor and the FOM. From our simulations, we find that the limitation of the maximal squeezing factor is about 46 with figure of merit > 1 , which is larger than that of *h*-BN-encapsulated graphene ($\beta_{max}/k_0 \sim 37$; *Science* **354**, aag1992 (2016)).

Figure R1. Design of a far-infrared type-I hyperbolic metasurface. (a) A unit cell of a far-infrared type-I hyperbolic metasurface. The yellow region is silver. Here, $w=0.1 \text{ }\mu\text{m}$; $a=0.2 \text{ }\mu\text{m}$; $n=7$; $p=2.9 \text{ }\mu\text{m}$; and the thickness of the silver is $0.1 \text{ }\mu\text{m}$. (b) Z-oriented magnetic field distributions at 2.65 THz. (c) Dispersion and FOM ($\text{Re}(k)/\text{Im}(k)$) of the designed far-infrared type-I hyperbolic metasurface, respectively. (d) Maximal squeezing factors and the corresponding FOM of the far-infrared type-I hyperbolic metasurfaces as a function of numbers of turns.

This discussion has been added in Supplementary Information together with the new section “S2. Design of a far-infrared type-I hyperbolic metasurface”.

Accordingly, in the main text, on page 13, starting from line 274, we have stated “Although demonstrated at microwave frequencies, the concept of type-I hyperbolic metasurface is general and applicable to higher frequencies such as the far-infrared regime (see a design of a far-infrared type-I hyperbolic metasurface in Supplementary Information).”

2nd Reviewer -- Comment 2:

On the same note, on line 213, you compare your high factor ranges from 8 to 60 to ordinary

metamaterial and metasurfaces. What are the usual squeezed factor ranges of ordinary metamaterial and metasurfaces? In the ref. 2 you cite, I don't see such discussion.

Response from Authors:

The effective refraction indexes (equivalent to the squeezing factors) in ordinary metamaterials, such as split ring resonators and negative-refraction metamaterials (e.g., $n_{eff} < 4$ in *Science* **306**, 1351, (2004); $n_{eff} < 4.5$ in *PRE* **70**, 016608 (2004)) and spoof plasmonic metamaterials (e.g., $\beta_{max}/k_0 < 1.1$ in *Nature Photon.* **2**, 175 (2008); $\beta_{max}/k_0 < 2.38$ in *PNAS* **110**, 40 (2013)), are usually less than 10. However, the effective refraction index can reach 38.6 in the specially designed terahertz metamaterials (*Nature* **369**, 7334 (2011)), which is comparable with ours.

To avoid the confusion, we have revised the statement on line 213. The new sentence on page 10, line 217, now reads

“The squeezing factor ranges from 8 up to 60, higher than typical values in many previous metamaterials and metasurfaces.^{11, 25, 30-32}”

2nd Reviewer -- Comment 3:

I am not an experimentalist and I think that the paper would be greatly improved with more details on the experimental section, discussed in line 200 to 208.

First, Fig. 4 needs more information. In Fig. 4a, the Hz field scans do not have x and y scale, so I don't understand if it is a scan of the total metasurface, only a unit cell, or X unit cells.

Response from Authors:

On the right side of Fig. 4a, we show the measured Hz field distributions over the whole metasurface which consists of 22×22 unit cells. Besides, we have added the scales in each subfigure.

To make it clear, we also have added a description in the main text, on page 10, starting from line 197, which reads

“The fabricated sample consists of 22×22 unit cells, taking up an area of 300 mm×300 mm (see Fig. 4(b)).”

2nd Reviewer -- Comment 4:

Looking at the dispersion, I would have expected a circular field pattern, but there seem to be two privileged directions (in the low frequency Hz scan): could you comment those results?

Response from Authors:

In the experiment, a point-like source is placed at the left boundary of the metasurface. At relatively high frequencies (from 0.213 GHz to 0.243 GHz), the field patterns are circular (see Fig. 4(a)). However, at relatively low frequencies, the field patterns show two privileged directions due to the “distorted” isofrequency contour near the Brillouin zone boundaries (see Fig. 1(b)).

To clarify this point, we have added a sentence in the main text, on page 11, starting from line 206, which reads

“Note that the field pattern at 0.198 GHz shows two privileged directions, due to the “distorted” isofrequency contour near the Brillouin zone boundaries (see Fig. 1(b)).”

2nd Reviewer -- Comment 5:

Also, are there reflections on the edge of the metasurface or is the designer polariton absorbed before reaching the edge?

Response from Authors:

Reflections occur at the metasurface edges, and we did not use absorbers to prevent them. In fact, such reflections are favorable and provide momenta with $k_x < 0$. This is why the measured isofrequency contours are almost circular.

To clarify this point, we have added a sentence in the main text, on page 10, starting from line 211, which reads

“One can see that the isofrequency contours are almost circular. This is because we don’t use the absorbers to prevent the reflections at the metasurface edges, which provide momenta of $k_x < 0$.”

2nd Reviewer -- Comment 6:

Or do we excite various frequencies with the incoming port? Could you describe in more details how you excite exactly the designer polariton?

Response from Authors:

In the experiment, the vector network analyzer (VNA) generates broadband signals with frequencies from 0^+ to 0.5 GHz. The incoming port directly connects to a coil unit cell at the metasurface edge, to drive the electrons in the coil, which is a high-efficiency way to excite the designer polaritons.

To clarify this point, we have added a sentence in the main text, on page 10, starting from line 198, which reads

“In the experiments, a port of vector network analyzer (VNA), as a broadband source (0^+ to 0.5 GHz), directly connects to a coil unit cell at the metasurface edge, to drive the electrons in the coil. This is a high-efficacy way to excite the designer polaritons in our system.”

2nd Reviewer -- Comment 7:

Is there a part of the excitation that is not coupled with it and creates radiative noise? Fig. 4b is too small to see whether it is fabricated or not, and there is no scale bar. Maybe you could provide a zoom of one unit cell?

Response from Authors:

Yes. There is a part of the excitation that creates radiative noise.

The wavevector of the radiative noise is above the light cone and, therefore, much smaller than that of the designer polariton. In the measured momentum space (see Fig. 4(a)), the spots at the first Brillouin zone center represent the radiative noise, while the circular patterns are the designer polariton modes. Compared with the designer polariton modes, the energy of the radiation noise is relatively small.

We have added a sentence in the main text, on page 10, starting from line 213, which reads

“Besides, the spots at FBZ center represent the radiative noise whose energy is relatively small compared with that of the designer polaritons.”

Figure 4(b) is the photograph of the fabricated sample, and we have added a scale bar in it. Besides, following the reviewer’s suggestion, we have provided a zoom of two unit cells of the fabricated sample in Fig. 4(b).

The updated Fig. 4 is shown below.

Updated Figure 4. Measured momentum space, field distributions in real space, and retrieved dispersion and group velocity of the designer polaritons on the type-I hyperbolic metasurface. (a) Measured momentum space of the designer polaritons in the FBZ, and measured Hz field distributions on the plane 5 mm over the metasurface (22 by 22 unit cells) at 0.198 GHz, 0.213 GHz, 0.228 GHz, and 0.243 GHz, respectively. The red star in the right inset represents the source location. (b) Photograph of the metasurface sample. Inset: two unit cells of the metasurface. (c) Retrieved squeezing factor and group velocity of the designer polaritons from the experimental results. Here, the blue dots are the experimental data; the blue curve represents exponential fits to the experimental data; the orange curve is the group velocity obtained from the blue curve.

2nd Reviewer -- Comment 8:

Typos:

- Lines 40 and 41: the “parallel” sign is not there.
- Line 49: change Especial to Special
- Subfigures of Fig. 3 are not lettered.
- Lines 175-176 : is the number of turns increased for fixed period?
- Figure 4 legend: orange curve of the group velocity, not yellow.
- Line 207, 5 nm or 2 mm?

Response from Authors:

We thank the reviewer for his/her carefulness. We have made the modifications accordingly.

Below are our replies to the 4th and 6th sub-comments, respectively.

Comment: Lines 175-176 : is the number of turns increased for fixed period?

Our reply: No. We alter the numbers of turns while maintain the other parameters (except the period) the same. Therefore, the period becomes longer as the number of turns increases.

Comment: 5 nm or 2 mm?

Our reply: 5 mm.

REVIEWERS' COMMENTS:

Reviewer #1 (Remarks to the Author):

I am satisfied with all the revisions and recommend publication of this manuscript.

Reviewer #2 (Remarks to the Author):

I thank the authors for their responses.

The responses on comments 3 to 7 on the experimental details satisfy me. I think that the clarifications on the figures and the added details in the main text will help the reader.

Following comment 1, I thank the authors for their time and the new results they provide in Supplementary Information. According to me, these simulations do not serve the argumentation of the authors, but show the limits of the design, since the FOM drops drastically for usual squeezed factors at those frequencies. Therefore, further developments are needed in order to access the optical and near-infrared ranges for the listed applications (Cherenkov, optical sensors,...).

However, the paper shows the proof of concept of a novel designer polariton in microwave and far-infrared range, by successfully combining simulations results with an effective model and experimental results. Thus, I still recommend the publication in Nature Communications. The only concern I have remains on listing applications in the introduction that do not work in the operating wavelength range. I would recommend rephrasing to the conditional tense, and adding a discussion at the end of the text about the limit of the design, and the future improvements.

Response Letter to Reviewers

We are grateful for the positive comments on this manuscript (NCOMMS-18-20953717B) from all the reviewers.

In the text below, each of the comments from each reviewer is quoted in *italics* and is followed by the corresponding detailed response. We have also revised the manuscript accordingly, and these updates are highlighted in **blue** and by a vertical **red** line in the left margin in those files. In the text below, the references to these updates are highlighted in a similar way.

COMMENTS FROM 1st REVIEWER:

I am satisfied with all the revisions and recommend publication of this manuscript.

Response from Authors:

We thank the reviewer for the positive comments and favorable recommendation.

COMMENTS FROM 2nd REVIEWER:

I thank the authors for their responses.

The responses on comments 3 to 7 on the experimental details satisfy me. I think that the clarifications on the figures and the added details in the main text will help the reader.

Following comment 1, I thank the authors for their time and the new results they provide in Supplementary Information. According to me, these simulations do not serve the argumentation of the authors, but show the limits of the design, since the FOM drops drastically for usual squeezed factors at those frequencies. Therefore, further developments are needed in order to access the optical and near-infrared ranges for the listed applications (Cherenkov, optical sensors,...).

However, the paper shows the proof of concept of a novel designer polariton in microwave and far-infrared range, by successfully combining simulations results with an effective model and experimental results. Thus, I still recommend the publication in Nature Communications. The only concern I have remains on listing applications in the introduction that do not work in the operating wavelength range. I would recommend rephrasing to the conditional tense, and adding a discussion at the end of the text about the limit of the design, and the future improvements.

Response from Authors:

We thank the reviewer for the positive comments and favorable recommendation.

We agree with the reviewer that directly extending the current design to optical and near-infrared regimes is challenging. However, the current design could work well from microwave up to terahertz frequencies, and find useful applications, such as terahertz sensing and low-electron-velocity terahertz Cherenkov radiation. Following the reviewer's suggestions, we have replaced "optical sensors" with "electromagnetic wave sensors" and "low-threshold Cherenkov radiation" with "low-threshold terahertz Cherenkov radiation" in our manuscript.

Besides, we have followed the reviewer's suggestion and added a discussion at the end of the main text about the limit of the design and the future improvements. It reads

"We also notice that at higher frequencies, metallic losses become considerable and cannot be neglected. Therefore, when increasing the maximal squeezing factor, the figure of merit

| $(\text{Re}(k)/\text{Im}(k))$ decreases, imposing a limit on the maximal squeezing factor. To overcome this challenge, one may use gain medium³⁸ to compensate for the dissipative losses, which needs future investigation.”